materials science/energy/crystallography

Prussian blue analogues, battery, storage, synthesis

**Author for correspondence:**
Anders Bentien
e-mail: bentien@eng.au.dk

This article has been edited by the Royal Society of Chemistry, including the commissioning, peer review process and editorial aspects up to the point of acceptance.

# Strategies for synthesis of Prussian blue analogues

Solveig Kjeldgaard[1,2], Iulian Dugulan[3], Aref Mamakhel[2], Marnix Wagemaker[3], Bo Brummerstedt Iversen[2] and Anders Bentien[1]

[1]Department of Engineering, and [2]Department of Chemistry, Aarhus University, Aarhus, Denmark
[3]Department of Radiation Science and Technology, Technical University Delft, Delft, The Netherlands

ID, 0000-0001-5559-9573; AB, 0000-0002-7204-9167

We report a comparison of different common synthetic strategies for preparation of Prussian blue analogues (PBA). PBA are promising as cathode material for a number of different battery types, including K-ion and Na-ion batteries with both aqueous and non-aqueous electrolytes. PBA exhibit a significant degree of structural variation. The structure of the PBA determines the electrochemical performance, and it is, therefore, important to understand how synthesis parameters affect the structure of the obtained product. PBA are often synthesized by co-precipitation of a metal salt and a hexacyanoferrate complex, and parameters such as concentration and oxidation state of the precursors, flow rate, temperature and additional salts can all potentially affect the structure of the product. Here, we report 12 different syntheses and compare the structure of the obtained PBA materials.

## 1. Introduction

Prussian blue analogues (PBA) are promising as cathode material for a number of different battery types because of their excellent redox properties and relatively high standard potential [1–7]. The cage-like structure exhibits wide channels, allowing for insertion of a wide range of intercalation ions. PBA can be prepared from abundant and non-toxic elements by simple and low-cost co-precipitation synthesis that is easily scalable. In addition to battery materials, PBA are promising as electrocatalysts for water splitting [8–13].

PBA have the general formula $A_x P[R(CN)_6]_{1-y} \cdot w H_2O$ where A is an insertion ion, often potassium or sodium, P and R are transition metals and $y$ is the number of $[R(CN)_6]$ vacancies. Cyanide groups connect transition metals P and R. The P atoms are coordinated to six nitrogen atoms, and the R atoms are coordinated to six carbon atoms, thus forming a framework with

**Figure 1.** Structure of PBA, $A_xP[R(CN)_6]_{1-y}$ with $y = 0$ (no vacancies), (a) cubic space group *Fm-3m* and (b) monoclinic space group $P2_1/c$. Water is omitted for clarity. Figures made using CrystalMaker.

large voids as shown in figure 1a. Vacancies are often present in PBA and is here defined as a lacking $[R(CN)_6]$ unit in the framework. The position itself is not vacant, but occupied by water molecules which may exist in three distinct configurations in the structure: zeolitic water positioned at the interstitial (A) sites, water coordinated to deficiently bonded metal ions at the vacancy sites, and water which is hydrogen-bonded to the coordinated water [14–16]. In the following, the abbreviation $A_xP[R]$ will be used for $A_xP[R(CN)_6]_{1-y} \cdot wH_2O$.

PBA offer many opportunities for structural variation and hence the properties are highly tunable. First of all, because PBA have an open framework structure with large interstitial sites, a wide range of different intercalation ions are possible, including alkali metals, divalent ions and even small molecules. The redox potential is affected by the insertion ion, and experiments show that the insertion potential of the alkali metals in PBA follow the trend $Li < Na < K$ [17–19]. Secondly, the transition metals on both the P and R site can be varied. Especially on the P site, many different transition metals have been explored, whereas the R site very often is occupied by iron. The redox potential of the R site metal is affected by the nature of the P site metal, with higher ionic potential (charge/radius) at the P site resulting in higher insertion potential [19]. Third, depending on synthesis conditions, PBA have a varying amount of structural vacancies. Many studies focus on minimizing the number of vacancies in PBA [20–28], while others suggest vacancies may play an important role in ion conduction and structural stability [4]. Whether a certain amount of vacancies is desirable or not may be influenced by the choice of intercalation ion and which electrolyte is used. Because water molecules finish the coordination sphere for P atoms at vacancies, more structural water is present for PBA with a higher number of vacancies. If using an organic electrolyte, the water has been shown to cause unwanted side reactions [21]. However, studies have also shown that larger insertion ions move via vacant sites [29,30], and a number of vacancies could, therefore, improve ion conduction. PBA have been used as cathode material also in aqueous cells, and here the structural water introduced with the vacancies does not cause side reactions, as water is already present.

Most PBA studies report cubic geometry (space group *Fm-3m*) [5,20–22,31,32], the unit cell of which is shown in figure 1a. Water molecules are omitted, and the figure shows a defect-free stoichiometry. In figure 1b, the monoclinic $P2_1/c$ structure is shown, where the transition metal octahedra are tilted and the $\beta$ angle is distorted. In this configuration, the $a$ and $b$ axes are reduced and the $c$ axis is expanded with respect to the cubic unit cell, resulting in a larger unit cell volume. In general, PBA with $x$ content above a critical level have rhombohedral [23–25,33] or monoclinic geometry [34,35]. The PBA structure can change during (de)intercalation of Na/K ions. Using operando PXRD on $Na_xFe[Fe]$, Brant et al. observed a phase transformation from rhombohedral at high sodium content to cubic at low sodium content [26]. Theoretical DFT studies confirm that the rhombohedral phase is most stable at full sodiation, whereas the cubic phase is most stable at full desodiation [36]. An exception is Mn[Fe] and Mn[Mn] where the most stable phase is tetragonal owing to the Jahn–Teller effect of manganese [36]. Experimentally, Mn[Mn] and Mn[Fe] are observed to be monoclinic at full sodiation [34,35]. Lee et al. [34] investigated Mn[Mn] and found phase transitions from cubic Mn [III][Mn [III]] (*Fm-3m*) to orthorhombic $Mn^{II}[Mn^{III}]$ ($P222_1$) to monoclinic $Mn^{II}[Mn^{II}]$ ($P2_1/n$) during reduction (ion insertion). Moritomo et al. synthesized $Na_xMn[Fe]_{1-y}$ with varying sodium content ($x$) and vacancies ($y$). For increasing $x$ and $y$, the initially cubic structure transitions to trigonal ($R$-$3c$) and then monoclinic ($P2/m$) phases [35]. In some cases, the crystal structure also depends on the intercalation

ion; $Na_xMn[Fe]$ can be cubic, monoclinic or rhombohedral depending on synthesis conditions and resulting sodium content, while $K_xMn[Fe]$ only exists in the monoclinic phase because of the larger ionic radius of potassium ions relative to sodium ions [27].

Prussian blue and its analogues have been synthesized in many different ways. Historically, Prussian blue pigments were synthesized using an organic precursor. The first synthesis method for preparation of Prussian blue was published in 1724 [37]. Herein, cattle blood was calcined with potassium hydrogen tartrate, alum and vitriol, and subsequently diluted with water and treated with hydrochloric acid. In the mid-nineteenth century, it became possible to synthesize potassium hexacyanoferrate in bulk [38], and after this point, Prussian blue could be synthesized from purely inorganic precursors. Nowadays, PBA are most often synthesized by co-precipitation of a metal salt and a hexacyanoferrate complex. This is an easy, low-cost and scalable synthesis. The metal salt is most often a sulfate, nitrate or chloride. The hexacyanoferrate complex can be $A_3Fe^{III}(CN)_6$ or $A_4Fe^{II}(CN)_6$ where A is potassium or sodium. The concentration of the solutions, flow rate, temperature and ageing are parameters that vary in different studies. Varying the synthesis parameters slightly can have a large impact on the obtained product.

Another common strategy for synthesis of Prussian blue is acid decomposition. Acid decomposition involved heating $K_4Fe(CN)_6$ in acid which will cause decomposition and free $Fe^{2+}$ which reacts with $K_4Fe(CN)_6$ to form Fe[Fe]. The acid decomposition gives a slower nucleation and grain growth, and, therefore, this type of synthesis usually results in a highly crystalline product with few vacancies. However, it can only be used to synthesize Fe[Fe], and it produces toxic byproducts in the form of ACN. This synthesis route is not explored in the current study.

Because the precipitation of PBA is very fast, and the solubility product of metal hexacyanoferrates is extremely low [39], Prussian blue and its analogues have a strong tendency to form small particles with a large number of $[Fe(CN)_6]^{3-/4-}$ vacancies. Both the number of vacancies and the particle size have been identified as key parameters for electrochemical performance of PBA, and it is, therefore, desirable to understand what influences these characteristics and be able to control these.

The size of PBA particles may be important to electrochemical performance. It has been shown that smaller particles result in higher capacities [7,40,41], which could be explained by the longer diffusion path for larger particles, or by a kinetic barrier for diffusion for larger particles [7]. Although the initial capacity is higher for small particles, electrochemical cycling of $K_xMn[Fe]$ revealed rapid capacity decay for small particles, and a longer cycle life for larger particles, with a particle size of approximately 200 nm being the best compromise between high capacity and long cycle life [7]. Particle size may be more important for $K_xP[R]$ than $Na_xP[R]$, as the diffusion of sodium ions is generally more facile than that of potassium ions.

In the current study, we report 12 syntheses of PBA prepared using common synthesis strategies and compare the obtained product. To enable comparison of different synthesis strategies, all PBA in the present study contain potassium as the intercalation ion, and the focus is on the hexacyanoferrates, R = Fe. In this work, $K_xP[Fe]$ have been prepared with P = Cu, Fe, Mn. Cu[Fe] are the promising cathode materials for aqueous batteries because of the relatively high redox potential just below the oxygen evolution potential. For Fe[Fe] and Mn[Fe], the potential is too high for aqueous electrolytes, and these should, therefore, be tested in an organic electrolyte or water-in-salt electrolyte. The present work focuses exclusively on structural characterization, while the electrochemical properties will be evaluated in a forthcoming study.

# 2. Experimental procedure

## 2.1. Synthesis

An overview of all syntheses is included in table 1. In syntheses A and B, a copper sulfate solution was added dropwise to $K_3Fe(CN)_6$ (sample A) or $K_4Fe(CN)_6$ (sample B) at room temperature. Syntheses C–L were prepared by co-precipitation using programmable syringe pumps (Aladdin). The two precursor solutions were pumped into a flask containing 50 ml of water placed in a water bath to control temperature. The flask was wrapped in metal foil to protect from too much light exposure. Following precipitation, the precipitate was aged in the mother liquor with heating and stirring on for another 2 h after which the heating was turned off and the solution was stirred for another 2 h at room temperature (referred to as 'ageing' in table 1). Finally, the precipitate was washed with water and dried. Parameters such as precursor concentration, temperature and flow rate are varied, and the

**Table 1.** Syntheses.

| sample | reagents | target oxidation state | conc (M) | flow rate (ml min$^{-1}$) | Atm | T (°C) | ageing |
|---|---|---|---|---|---|---|---|
| A | $CuSO_4$ | Cu(II) | 0.15 | — | | — | RT | — |
| | $K_3Fe(CN)_6$ | Fe(III) | 0.05 | | | | |
| B | $CuSO_4$ | Cu(II) | 0.15 | — | | — | RT | — |
| | $K_4Fe(CN)_6$ | Fe(II) | 0.05 | | | | |
| C | $CuSO_4$ | Cu(II) | 0.1 | 4 | $N_2$ | 70 | 2 h heat, |
| | $K_3Fe(CN)_6$ | Fe(III) | 0.1 | | | | 2 h stir |
| D | $CuSO_4$ | Cu(II) | 0.1 | 4 | $N_2$ | 70 | 2 h heat, |
| | $K_4Fe(CN)_6$ | Fe(II) | 0.1 | | | | 2 h stir |
| E | $FeSO_4$ | Fe(II) | 0.1 | 4 | $N_2$ | 70 | 2 h heat, |
| | $K_3Fe(CN)_6$ | Fe(III) | 0.1 | | | | 2 h stir |
| F | $FeSO_4$ | Fe(II) | 0.1 | 4 | $N_2$ | 70 | 2 h heat, |
| | $K_4Fe(CN)_6$ | Fe(II) | 0.1 | | | | 2 h stir |
| G | $CuSO_4$ | Cu(II) | 0.1 | 4 | — | 70 | 2 h heat, |
| | $K_3Fe(CN)_6$ | Fe(III) | 0.1 | | | | 2 h stir |
| | KCl | | 2.5 | | | | |
| H | $FeCl_3$ | Fe(III) | 0.1 | 4 | — | 70 | 2 h heat, |
| | $K_3Fe(CN)_6$ | Fe(III) | 0.1 | | | | 2 h stir |
| | KCl | | 2.5 | | | | |
| I | $CuSO_4$ | Cu(II) | 0.1 | 4 | — | 70 | 2 h heat, |
| | $K_3Fe(CN)_6$ | Fe(III) | 0.15 | | | | 2 h stir |
| J | $CuCl_2$ | Cu(II) | 0.1 | 1 | $N_2$ | RT | 18 h stir |
| | $K_4Fe(CN)_6$ | Fe(II) | 0.1 | | | | |
| K | $CuCl_2$ | Cu(II) | 0.1 | 1 | $N_2$ | RT | 18 h stir, |
| | $K_4Fe(CN)_6$ | Fe(II) | 0.1 | | | | $H_2O_2$ |
| L | $MnCl_2$ | Mn(II) | 0.125 | 1 | $N_2$ | 70 | 0.5 h |
| | $K_3C_6H_5O_7$ | Fe(II) | 1.25 | | | | heat + stir |
| | $K_4Fe(CN)_6$ | | 0.125 | | | | |
| | KCl | | 1.5 | | | | |

relevant parameters for each synthesis can be found in table 1. Syntheses C–F and J–L were purged with nitrogen throughout the synthesis to avoid oxidation. For syntheses G and H, KCl was added to the 50 ml of water, making a 2.5 M solution. Samples J and K are from the same synthesis batch. Following precipitation and ageing, the batch was split in two, and K was acidified with HCl and oxidized using hydrogen peroxide. In synthesis L, potassium citrate was added to the manganese chloride solution, and potassium chloride was added to the potassium hexacyanoferrate solution. Syntheses G, H and I were performed at the Technological University Delft (The Netherlands), all others at Aarhus University (Denmark). The syringe pumps and set-up were the same in both laboratories, except the solutions were not purged with nitrogen at TU Delft.

## 2.2. Materials characterization

### 2.2.1. Elemental composition

Elemental composition was determined by inductively coupled plasma optical emission spectrometry (ICP-OES) using a SPECTRO ARCOS instrument from AMETEK. PBA were dissolved in aqua regia and heated in an autoclave for 2 h at 150°C.

### 2.2.2. Electron microscopy

Transmission electron microscopy (TEM) images were obtained on a TALOS F200A with a TWIN lens system, X-FEG electron source, Ceta 16 M camera and a Super-X EDS detector. Spatially resolved elemental analysis with a spatial resolution better than 2 nm was obtained using the same TALOS microscope in STEM mode. STEM pictures were obtained using a high angle annular dark field detector (HAADF).

### 2.2.3. Powder X-ray diffraction

Powder X-ray diffraction (PXRD) measurement of samples C–I was performed using a PANalytical X'Pert Pro PW3040/60 diffractometer with Cu K$\alpha$ radiation operating at 45 kV and 40 mA with an angular range of $2\theta = 10$–$100°$. PXRD measurement of samples A, B and J–L was performed using a Rigaku SmartLab diffractometer with Cu K$\alpha$ radiation operating at 180 mA and 40 kV with an angular range of $2\theta = 10$–$90°$.

Data analysis was carried out using Rietveld refinement [42] implemented in *FullProf* [43]. The initial structural model for samples A–K was taken from ICSD-89338, and for sample L, the initial structural model was adapted from Fiore *et al.* [44]. Refined parameters include zero-point displacement, scale factor, lattice parameters, background (linear interpolation between a set of background points with refinable height, manually chosen to fit features in the background without affecting the Bragg peaks), peak-profile parameters (X, Y, U and IG from the Thompson–Cox–Hastings pseudo-Voigt model [45]). Peak-profile parameters that obtained very small values when refined were fixed to zero. Transition metal occupancies were not refined, but fixed to the values obtained from ICP-OES owing to similar scattering powers. An instrument resolution file determined by measuring data on a LaB$_6$ standard in the same experimental configuration was used to account for instrumental broadening.

### 2.2.4. Mössbauer spectroscopy

Transmission $^{57}$Fe Mössbauer absorption spectra were collected at 300 and 4.2 K with a conventional constant-acceleration or sinusoidal velocity spectrometers using a $^{57}$Co(Rh) source. Velocity calibration was carried out using an α-Fe foil at room temperature. The Mössbauer spectra were fitted using the MossWinn 4.0 program [46].

## 3. Results and discussion

Table 1 lists the 12 different syntheses, and variation of synthesis parameters.

The stoichiometry as determined by ICP-OES and Mössbauer spectroscopy is reported in table 2. Fitted parameters from Mössbauer spectroscopy are found in the electronic supplementary material. The stoichiometry of copper hexacyanoferrates and manganese hexacyanoferrate were determined by ICP-OES. From ICP, the K : P : Fe ratio is given. Normalizing to [P] = 1, the stoichiometry K$_x$P[Fe(CN)$_6$]$_{1-y}$ can be determined, as the iron deficiency corresponds to the number of [Fe(CN)$_6$] vacancies. For samples with P = Fe, the P : R ratio cannot be determined by ICP-OES because both sites are occupied by iron. In this case, Mössbauer spectroscopy can be used to investigate the iron environment. Iron coordinated to six carbon atoms experience a strong ligand field and are low spin (LS), whereas iron coordinated to six nitrogen atoms experience a weak ligand field and are high spin (HS). From the LS/HS ratio, the stoichiometry is determined. The potassium content in iron hexacyanoferrates (P = R = Fe) is determined by charge balancing. An example of how stoichiometry is calculated for Fe[Fe] samples is provided in electronic supplementary material.

PBA are often prepared by co-precipitation, which has the advantage of keeping concentrations constant. A more simple strategy is to add one solution to the other. Samples A and B were prepared by dropwise addition of copper sulfate to the potassium hexacyanoferrate solution. Compared to the other syntheses, which are all variations on co-precipitation, samples A and B have a higher number of [Fe(CN)$_6$]$^{3-/4-}$ vacancies. Most of the samples have a vacancy concentration close to 1/3, which seems to be strongly favoured, despite different synthesis conditions. The stoichiometry P[II][R$^{III}$(CN)$_6$]$_{2/3}$ is often reported in the literature [47,48], where intercalation ions are absent and 1/3 of [R(CN)$_6$]$^{3-}$ sites are vacant in order for charge balance of the framework to be fulfilled. In synthesis L, potassium citrate is added to the metal salt precursor solution. For sample L, $y = 0.04$, which is a much lower number of vacancies than any of the other syntheses. Citrate is effective for

**Table 2.** Stoichiometry by Mössbauer spectroscopy and ICP-OES.

| sample | Mössbauer | | | ICP-OES |
| | phase | spectral contribution (%) | stoichiometry | stoichiometry |
| --- | --- | --- | --- | --- |
| A | — | — | — | $K_0Cu[Fe(CN)_6]_{0.59}$ |
| B | — | — | — | $K_{0.23}Cu[Fe(CN)_6]_{0.49}$ |
| C | C–Fe$^{II}$ | 34 | — | $K_{0.24}Cu[Fe(CN)_6]_{0.69}$ |
| | C–Fe$^{III}$ | 66 | | |
| D | C–Fe$^{II}$ | 100 | — | $K_{1.10}Cu[Fe(CN)_6]_{0.73}$ |
| E | C–Fe$^{II}$ | 50 | $KFe^{III}Fe^{II}(CN)_6$ | — |
| | N–Fe$^{III}$ | 50 | | |
| F | C–Fe$^{II}$ | 44 | $K_{0.25}Fe^{III/II}[Fe^{II}(CN)_6]_{0.79}$ | — |
| | N–Fe$^{III}$ | 51 | | |
| | N–Fe$^{II}$ | 5 | | |
| G | C–Fe$^{II}$ | 11 | — | $K_{0.18}Cu[Fe(CN)_6]_{0.69}$ |
| | C–Fe$^{III}$ | 89 | | |
| H | C–Fe$^{II}$ | 15 | $K_{0.74}Fe^{III}[Fe^{III/II}(CN)_6]_{0.67}$ | — |
| | C–Fe$^{III}$ | 25 | | |
| | N–Fe$^{III}$ | 60 | | |
| I | C–Fe$^{II}$ | 37 | — | $K_{0.10}Cu[Fe(CN)_6]_{0.68}$ |
| | C–Fe$^{III}$ | 63 | | |
| J | — | — | — | $K_{1.51}Cu[Fe(CN)_6]_{0.79}$ |
| K | — | — | — | $K_{0.86}Cu[Fe(CN)_6]_{0.72}$ |
| L | — | — | — | $K_{1.86}Mn[Fe(CN)_6]_{0.97}$ |

controlling the stoichiometry because citrate chelates with the metal salt and the metal–citrate chelate reacts slower with the hexacyanoferrate complex, effectively slowing down nucleation and growth of PBA. The addition of sodium citrate and potassium citrate has previously been reported to successfully prepare low vacancy PBA [22–24,27,40,44,49–51].

Owing to the fast precipitation kinetics, PBA particles are generally very small, often within the range 10–50 nm. In figure 2, TEM images of samples A to L are shown. The particle size as determined from TEM is reported in table 3. Also included in table 3 is the particle size determined from PXRD. PXRD gives an average size of coherently scattering domains within a grain, and may, therefore, differ from the particle size as observed from electron microscopy, as these may be agglomerates of smaller crystallites or a mixture of amorphous and crystalline material.

The largest particles (PXRD) are present in sample H and sample L. From PXRD, the average particle size of sample L is 116 nm. The particles have a wide size distribution and an undefined shape. The addition of citrate has been shown to increase the particle size of PBA, and the higher the citrate to $P^{2+}$ ratio is, the larger is the increase in particle size [7,51].

Samples G and H are prepared in the presence of excess potassium chloride. The largest particles are obtained for sample H, where the average particle size (PXRD) is 192 nm, which is much larger than the approximately 10–40 nm that is seen for PBA synthesized without addition of potassium chloride. The particle size distribution is relatively narrow, and the particles have a cubic morphology. Sample H is Prussian green, $Fe^{III}[Fe^{III}]$, which, as the name indicates, has an intense green colour. The colour appeared only slowly indicating the crystallization was slowed owing to the addition of KCl. Sample G is Cu[Fe]. Unlike sample H, the particle size distribution of G is very broad. From TEM images, it can be seen that there are two kinds of particles present; larger cubic particles and smaller round particles. Sample G is prepared similarly to sample H, but whereas the metal salt is $CuSO_4$ in synthesis G, it is $FeCl_3$ in synthesis H. It is unknown what exactly causes the difference in particle size between samples G and H. Samples J and K are synthesized using a metal chloride precursor as well, and these samples both have a small particle size (11 and 32 nm), so the increase in particle size cannot be explained by using a metal chloride salt.

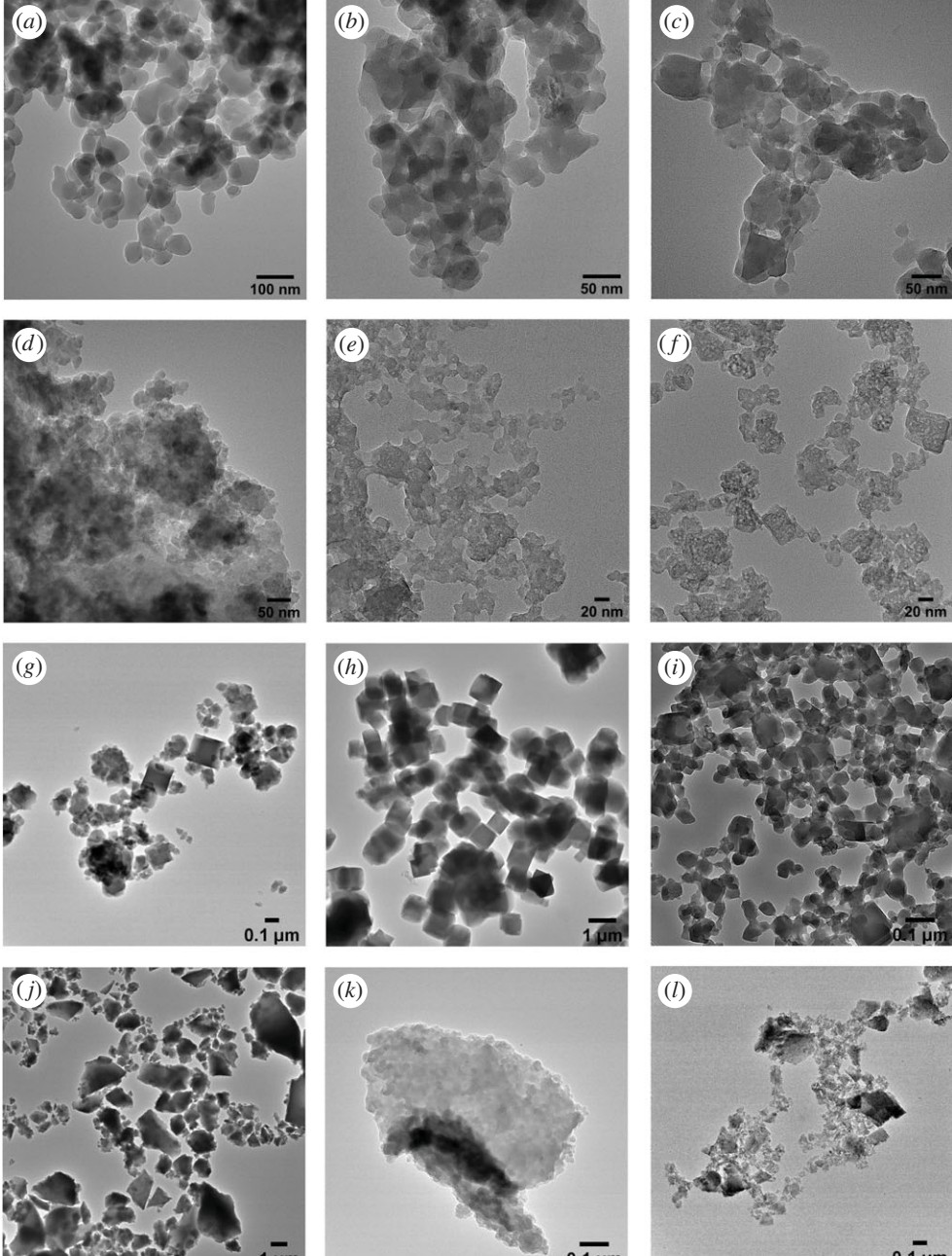

**Figure 2.** Bright field TEM images.

In the literature, the addition of KCl (or NaCl) has been reported to increase the cation content in PBA [35], decrease the vacancy concentration [34,35] and decrease the particle size distribution [52]. A recent study reporting a number of syntheses with/without KCl and varying citrate concentration reports that the addition of KCl decreases particle size, and has no significant influence on the number of $Fe(CN)_6$ vacancies [44]. Li *et al.* [53] have reported decreased particle size with increasing precursor concentration, and by increased flow rate.

Particle agglomeration is seen for samples J and K, and to some extent in sample L. For samples J and K, the agglomeration is so pronounced that it is difficult to distinguish individual particles in TEM images. For syntheses J, K and L, the flow rate was $1\,ml\,min^{-1}$, whereas the flow rate for syntheses C–I was $4\,ml\,min^{-1}$. This may indicate that a very slow flow rate could increase particle agglomeration. Samples J and K were prepared at room temperature, but as samples A and B, which were also prepared at room temperature, show no significant agglomeration, the agglomeration does not seem to be caused by low temperature.

**Table 3.** Crystallite and particle size as determined by PXRD and TEM, respectively.

| sample | stoichiometry | PXRD | | TEM |
| --- | --- | --- | --- | --- |
| | | size (nm) | strain (%) | mean size (s.d.) (nm) |
| A | $K_0Cu[Fe(CN)_6]_{0.59}$ | 49.5 | 9.45 | 55 (16) |
| B | $K_{0.23}Cu[Fe(CN)_6]_{0.49}$ | 23.6 | 42.70 | 30 (9) |
| C | $K_{0.24}Cu[Fe(CN)_6]_{0.69}$ | 33.0 | 36.27 | 30 (8) |
| D | $K_{1.10}Cu[Fe(CN)_6]_{0.73}$ | 14.5 | 61.59 | 20 (4) |
| E | $KFeFe(CN)_6$ | 25.3 | 134.31 | 15 (3) |
| F | $K_{0.25}Fe[Fe(CN)_6]_{0.79}$ | 16.4 | 29.41 | 20 (5) |
| G | $K_{0.18}Cu[Fe(CN)_6]_{0.69}$ | 53.3 | 25.66 | 80 (40) |
| H | $K_{0.74}Fe^{III}[Fe^{III/II}(CN)_6]_{0.67}$ | 192.4 | 33.60 | 800 (120) |
| I | $K_{0.10}Cu[Fe(CN)_6]_{0.68}$ | 37.3 | 27.69 | 48 (18) |
| J | $K_{1.51}Cu[Fe(CN)_6]_{0.79}$ | 11.2 | 98.08 | a |
| K | $K_{0.86}Cu[Fe(CN)_6]_{0.72}$ | 31.9 | 54.40 | 24 (5) |
| L | $K_{1.86}Mn[Fe(CN)_6]_{0.97}$ | 116.1 | 11.06 | 71 (49) |

[a]Individual particle size cannot be determined owing to agglomeration.

The diffractograms of samples A–L and Bragg indices for space groups $Fm\text{-}3m$ and $P2_1/c$ are shown in figure 3a. To better see peak shift and broadening, a zoom of samples A–K ($2\theta = 16\text{–}28°$) is shown in figure 3b. Lattice parameters and atomic displacement parameters (ADPs) determined by Rietveld refinement are listed in table 4. Rietveld refinement fits and further crystallographic details are found in the electronic supplementary material.

## 3.1. Oxidized/reduced form

PBA can be easily synthesized in both the oxidized and reduced form, as metal salts and potassium hexacyanoferrate are available in different oxidation states. Samples A/B and C/D and can be compared in pairs because they are synthesized under exactly the same conditions, except A and C are made from $K_3Fe(CN)_6$, whereas B and D are made from $K_4Fe(CN)_6$. Similarly, J and K can be compared, although here the samples are from the same synthesis batch and K is subsequently oxidized.

Bragg peaks for $Cu^{II}[Fe^{II}]$ synthesized in the reduced form (samples B, D, J) are shifted to higher $2\theta$ compared to $Cu^{II}[Fe^{III}]$ synthesized in the oxidized form (samples A, C, G, I), indicating smaller unit cell parameters for the reduced form. It may be a little counterintuitive that the reduced form, which contains more potassium, should decrease lattice parameters; however, the radius of $[Fe(CN)_6]^{4-}$ is decreased compared to $[Fe(CN)_6]^{3-}$ owing to an increase in π back-bonding of Fe(II) to carbon [54]. It is well established in the literature that reduction in PBA causes a decrease in lattice parameters [4,5,55]. J and K have very similar lattice parameters, the lattice parameters of K are only very slightly increased with oxidation. $Cu^{II}[Fe^{III}]$ synthesized in the oxidized form (A, C, G, I, K) have sharper peaks than $Cu^{II}[Fe^{II}]$ synthesized in the reduced form (B, D, J). The peak broadening of reduced samples (B, D, J) is owing to a combination of decreased particle size and increased strain. The lattice parameters of samples E and F are very similar. From Mössbauer spectroscopy, the $Fe^{2+}/Fe^{3+}$ ratio is 50/50 in sample E and 49/51 in sample F, and the similar $Fe^{2+}/Fe^{3+}$ ratios may explain the similar unit cell size (figure 4).

Fe[Fe] samples (E, F, H) are shifted to lower $2\theta$ compared to the Cu[Fe] samples. The lattice parameters increase with the size of the $P^{2+}$ ion, and because $Fe^{2+}$ is larger than $Cu^{2+}$, the lattice parameters for Fe[Fe] samples are larger than those of Cu[Fe]. $Mn^{2+}$ is even larger, and Mn[Fe] is often reported having a monoclinic structure. The monoclinic unit cell is larger than the cubic unit cell, explaining the tendency of Mn[Fe] to have the monoclinic structure. It has been argued that Mn[Fe] will always be monoclinic owing to the large size of $Mn^{2+}$ [56]; however, also Mn[Fe] has been reported in cubic form [57], as well as monoclinic Cu[Fe] has been prepared [58].

For several of the samples, the ADPs of K refines to unphysical values, and is, therefore, fixed to 0.5. The large ADPs might suggest disorder, or it could be a reflection of the large number of

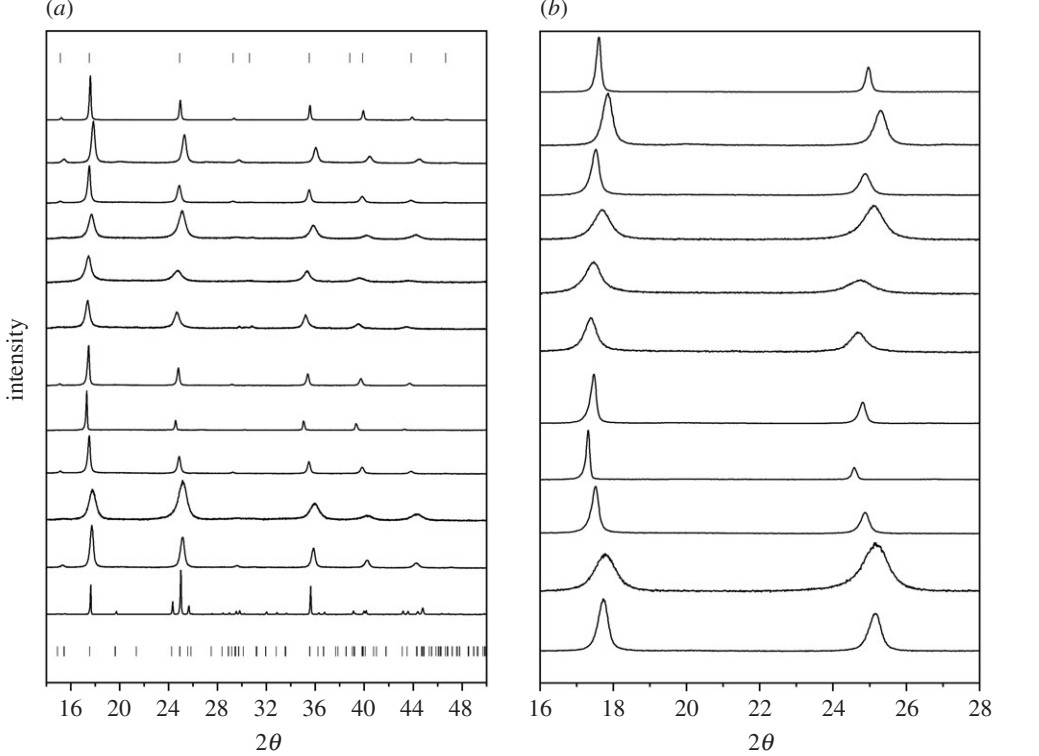

**Figure 3.** (a) PXRD samples A–L (top to bottom) with Bragg indices for space group *Fm-3m* (top) and *P2₁/c* (bottom). (b) Zooming in on $2\theta = 16$–$28°$ to better see peak shifts and broadening (sample L excluded).

**Table 4.** Rietveld refined parameters.

| sample | stoichiometry | unit cell (Å) | atomic displacement (Å$^2$) | | | | |
|---|---|---|---|---|---|---|---|
| | | | K | P | R | C | N |
| A | $K_0Cu[Fe(CN)_6]_{0.59}$ | 10.1030(1) | — | 3.0 | 0.31 | 7.3 | 0.79 |
| B | $K_{0.23}Cu[Fe(CN)_6]_{0.49}$ | 9.9877(4) | 0.5$^a$ | 4.3 | $^a$ | 8.8 | 2.3 |
| C | $K_{0.24}Cu[Fe(CN)_6]_{0.69}$ | 10.1007(2) | 0.5$^a$ | 0.83 | 0.33 | 4.9 | 2.2 |
| D | $K_{1.10}Cu[Fe(CN)_6]_{0.73}$ | 10.0215(3) | 3.1 | 1.9 | 3.9 | 8.3 | 2.1 |
| E | $KFeFe(CN)_6$ | 10.1691(7) | $^a$ | $^a$ | $^a$ | $^a$ | $^a$ |
| F | $K_{0.25}Fe[Fe(CN)_6]_{0.79}$ | 10.1820(6) | $^a$ | $^a$ | $^a$ | $^a$ | $^a$ |
| G | $K_{0.18}Cu[Fe(CN)_6]_{0.69}$ | 10.1279(2) | 0.5$^a$ | 1.2 | 0.24 | 4.7 | 0.67 |
| H | $K_{0.74}Fe^{III}[Fe^{III/II}(CN)_6]_{0.67}$ | 10.2227(3) | $^a$ | $^a$ | $^a$ | $^a$ | $^a$ |
| I | $K_{0.10}Cu[Fe(CN)_6]_{0.68}$ | 10.1038(2) | 0.5$^a$ | 1.1 | 0.15 | 3.3 | 4.0 |
| J | $K_{1.51}Cu[Fe(CN)_6]_{0.79}$ | 10.0188(6) | 2.5 | 2.9 | 2.5 | 4.5 | 6.9 |
| K | $K_{0.86}Cu[Fe(CN)_6]_{0.72}$ | 10.0264(2) | 5.6 | 1.3 | 1.7 | 6.3 | 0.85 |
| L | $K_{1.86}Mn[Fe(CN)_6]_{0.97}$ | *a*: 6.9563(7) | 4.0 | 1.5 | 1.6 | C1: 0.73 | N1: 1.3 |
| | | *b*: 7.3279(7) | | | | C2: 2.3 | N2: 0.21 |
| | | *c*: 12.248(1) | | | | C3: 0.57 | N3: 2.0 |

$^a$Fixed.

$[Fe(CN)_6]^{3-/4-}$ vacancies. ADPs cannot be refined for Fe[Fe] samples (E, F and H). The intensities are not very well described for the Fe[Fe] samples, which could be owing to the occupancies being off. For the Fe[Fe] samples, the occupancy of P and R site is determined by Mössbauer spectroscopy, whereas ICP-

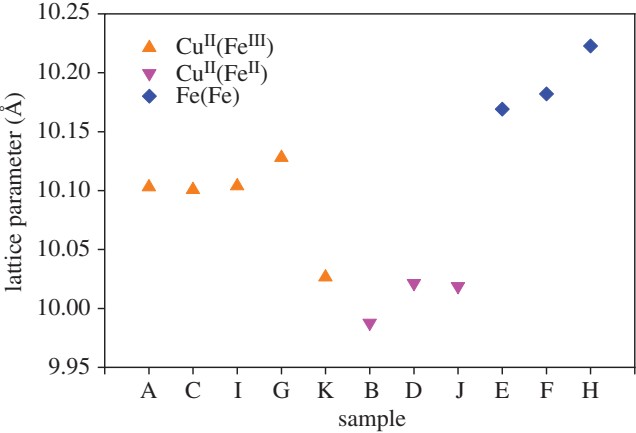

**Figure 4.** Unit cell parameters for the different PBA samples.

OES is used to determine occupancy for all other samples. It is, therefore, believed that the occupancies are not entirely well described from Mössbauer spectroscopy. Although the presence of interstitial water in PBA is well established, water molecules are not included in refinement. Including water molecules in the fit could probably improve the fit.

## 3.2. Atmosphere

Syntheses C–F and J–L were purged with nitrogen throughout the synthesis to avoid oxidation of the product. It is well known that Prussian white, $Fe^{II}[Fe^{II}]$, easily oxidizes to Prussian blue, $Fe^{III}[Fe^{II}]$, which is directly visible as a colour change from white (or light blue) to the characteristic deep blue of Prussian blue. In synthesis F, Prussian white is prepared. The precipitate is initially light blue. When the flask is opened to the surrounding atmosphere and the powder is washed, a colour change is observed, and afterwards, the washed and dried powder cannot be visible distinguished from sample E, Prussian blue. In order to keep Prussian white in the reduced form, it must be stored in an inert atmosphere.

PBA synthesized in the oxidized state are often partially reduced. The oxidation potential of the PBA is often so high that it can be somewhat reduced by water [59]. Mössbauer spectroscopy reveals partial reduction of as-synthesized $Cu^{II}[Fe^{III}]$: in sample C, 34% of iron has been reduced to iron(II), in sample G, 11% of iron has been reduced to iron(II) and in sample I, 37% of iron has been reduced to iron(II). Purging with nitrogen makes no discernible difference for reduction of iron. Synthesis C is purged with nitrogen, and syntheses G and I are not. The amount of iron that has been reduced is similar for C and I, and for synthesis G, it is less. In synthesis G, KCl was added, which may be why less iron has been reduced.

## 3.3. Precursor concentration

Samples C and I are prepared very similarly, except sample I is made with excess $K_3Fe(CN)_6$, whereas sample C is made with equimolar amounts of $CuSO_4$ and $K_3Fe(CN)_6$. Since PBA have a tendency to form $Fe(CN)_6$ vacancies, one might speculate that increasing the $K_3Fe(CN)_6$ concentration could lower the amount of vacancies. In fact, C and I have very similar stoichiometry; a low potassium content of 0.24 (C) and 0.10 (I), and $[Fe(CN)_6]^{3-}$ vacancies are 0.31 (C) and 0.32 (I). From Mössbauer spectroscopy, 34% of iron is Fe(II) for C and 37% for I. The higher proportion of Fe(II) in sample I correlates well with a higher potassium content for sample C in terms of charge balance. Sample C was bubbled with $N_2$ during synthesis, which could explain the slightly lower oxidation and higher potassium content. Although the average size obtained from PXRD is very similar, the particle size distribution is wider for sample I, where TEM reveals particles ranging from 26 to 129 nm, whereas sample C has particles from 20 to 60 nm. Some particles in sample I have cubic morphology; in sample C, particles are round. The difference in size distribution is more likely owing to the different flow rate, 4 ml min$^{-1}$ for synthesis C and 1 ml min$^{-1}$ for synthesis I. It does not seem to be of major importance whether $K_3Fe(CN)_6$ is in excess or balanced to $CuSO_4$.

Samples A and B were prepared in the presence of excess $Cu^{2+}$. A higher concentration of the $P^{2+}$ is a strategy that has been used by Cui and co-workers [3,5,60], and they have argued that excess $P^{2+}$ slows down the crystallization, improving the quality of the product. Syntheses A and B are not prepared by

co-precipitation like the rest of the syntheses, and it is, therefore, difficult to evaluate the effect of excess $Cu^{2+}$ from these data. Most of the syntheses in the current study are done with equimolar amounts of $P^{2+}$ and hexacyanoferrate. In the literature, different approaches have been taken, including equal volume and a higher concentration of $P^{2+}$ [5,23], equal concentration but larger volume of $P^{2+}$ [20], and equal volumes and a higher concentration of hexacyanoferrate [22,24,32].

## 3.4. Metal salt

In syntheses J and D, $Cu^{II}[Fe^{II}]$ is prepared. Samples J and D are very similar in terms of stoichiometry; the potassium content in J is a little higher and the number of vacancies a little lower than for sample D. The average particle size of samples J and D is also similar, but in sample J, the particles agglomerate. The samples are synthesized differently; sample J is prepared from $CuCl_2$, whereas sample D is prepared from $CuSO_4$, the flow rate in synthesis J is lower, and the synthesis is done at room temperature.

Sample H is also prepared from a metal chloride, and here, no particle agglomeration is observed. It is, therefore, more likely that the particle agglomeration seen for sample J is caused by low synthesis temperature or low flow rate than the metal salt being a chloride. In sample L, the particles also agglomerate. Synthesis L was done with a low flow rate and a temperature of 70°C. It indicates that the low flow rate causes particle agglomeration. Unfortunately, flow rate is often not reported in the literature, and it is, therefore, difficult to compare to previous studies.

## 3.5. Temperature

Precipitation of PBA is commonly performed at relatively low temperatures, usually in the range from room temperature (RT) to approximately 80°C. Previous studies have reported a decreasing number of vacancies with lower temperature [21]. Syntheses using citrate as chelating agent observe the opposite trend; lower vacancy content and larger particles when increasing the temperature [27]. The difference may be owing to the very good chelating ability of the citrate; slow release of $P^{2+}$ owing to citrate coupled with higher diffusion of potassium and $[Fe(CN)_6]^{4-}$ owing to increased temperature yields a very low vacancy product.

# 4. Conclusion

In this study, we have investigated several common synthesis strategies for the preparation of PBA and compared the obtained product. Except sample L, all samples have cubic geometry and belong to the space group *Fm-3m*. Sample L, $K_{1.86}Mn[Fe(CN)_6]_{0.97}$, has a high potassium content and very low number of vacancies, which leads to a distortion of the cubic framework to a monoclinic ($P2_1/n$) structure which is associated with an increase in the unit cell volume.

PBA are prepared by mixing a metal salt and a hexacyanoferrate complex, either by simply adding one solution to the other or by co-precipitation. Co-precipitation synthesis has the advantage of keeping the concentration of reagents constant. Samples A and B, which were prepared by adding $CuSO_4$ to the hexacyanoferrate solution, have a higher number of $[Fe(CN)_6]^{3-/4-}$ vacancies than all other samples. Most of the samples have a vacancy concentration close to 1/3, which seems to be strongly favoured, despite different synthesis conditions. The rapid precipitation of PBA lead to vacancies, and, therefore, an effective way to limit the number of $[Fe(CN)_6]^{3-/4-}$ vacancies is to slow down nucleation and growth. As demonstrated by synthesis L, the most effective strategy for controlling stoichiometry is using a citrate chelate, where a very low number of vacancies, $y = 0.04$, was achieved.

Most syntheses result in small particles of approximately 10–50 nm in size. The addition of citrate increased the particle size considerably, resulting in an average particle size of 116 nm. However, the largest average particle size is reported for sample H, $Fe^{III}[Fe^{III}]$ prepared from $FeCl_3$ and $K_3Fe(CN)_6$ in the presence of excess KCl, with an average particle size of 192 nm. Furthermore, sample H exhibited particles particles with uniform size and a cubic morphology. Synthesis of $Cu^{II}[Fe^{III}]$ in excess KCl yielded particles with an average size of 53 nm, which is larger than syntheses without KCl, but considerably less than $Fe^{III}[Fe^{III}]$ prepared with excess KCl. Particle agglomeration is observed in samples J, K and L which were synthesized using a lower flow rate of 1 ml min$^{-1}$.

PBA were prepared in both the oxidized and reduced form. Bragg peaks are broader for Cu[Fe] samples synthesized in the reduced form compared to Cu[Fe] samples synthesized in the oxidized form owing to a combination of a smaller particle size and increased strain. Peaks are moved to

higher angle for Cu[Fe] in the reduced form, indicating decreased lattice parameters, as the radius of $[Fe(CN)_6]^{4-}$ is decreased compared to $[Fe(CN)_6]^{3-}$. For several of the samples, Rietveld refinement of ADPs resulted in unphysically large values. The large ADPs might suggest disorder, or it could be a reflection of the large number of $[Fe(CN)_6]^{3-/4-}$ vacancies present, and it may be interesting to investigate further.

Data accessibility. The datasets supporting this article have been uploaded as part of the electronic supplementary material.

Authors' contributions. S.K. carried out the laboratory work with respect to synthesis and characterization. I.D. carried out Mössbauer spectroscopy, while A.M. performed electron microscopy. S.K., M.W., B.B.I. and A.B. conceived the study, designed the study, coordinated the study and helped draft the manuscript. All authors gave final approval for publication and agree to be held accountable for the work performed therein.

Competing interests. We declare we have no competing interests.

Funding. The Danish Research Council for Independent Research (grant no. DFF–4005-00517) and iMAT-Aarhus University Centre for Integrated Materials Research are acknowledged for financial funding.

Acknowledgements. Martin Bondesgaard is gratefully acknowledged for running ICP-OES on samples A and B.

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
