## [Reviewer comments · Royal Society Open Science]

Review History

RSOS-201779.R0 (Original submission)

Review form: Reviewer 1

Is the manuscript scientifically sound in its present form?

Yes

Are the interpretations and conclusions justified by the results?

Yes

Is the language acceptable?

Yes

Do you have any ethical concerns with this paper?

No

Have you any concerns about statistical analyses in this paper?

No

Recommendation?

Accept with minor revision (please list in comments)

Comments to the Author(s)

The paper entitled "Strategies for Synthesis of Prussian Blue Analogues" describes different syntheses and compares the structure of the obtained PBA materials. The PBAs are well characterized by PXRD, Mössbauer spectroscopy, and TEM studies. The manuscript is well organized and uncovered the properties of different PBAs. Therefore, the manuscript can be accepted for publication after minor revision.

The critical comments are as follow:

1. Grammatical errors should be checked again.
2. IR/Raman spectroscopy could be an important tool to understand the bridging of -CN group.
3. The PBAs prepared with potassium tetracyanonickelate should be discussed in the introduction.
4. Recent references for the application of PBAs (Mater. Today Energy 2020, 16, 100404, Chem. Asian J. 2020, 15, 607-623) should be included in the introduction part.
5. Author should describe how the tuneable structure of PBAs affects their electrochemical performance.
6. Author should explain the effect of the insertion of various metal ions at P and R sites in different PBAs.

Review form: Reviewer 2

Is the manuscript scientifically sound in its present form?

No

Are the interpretations and conclusions justified by the results?

Yes

Is the language acceptable?

Yes

Do you have any ethical concerns with this paper?

No

Have you any concerns about statistical analyses in this paper?

No

Recommendation?

Accept with minor revision (please list in comments)

Comments to the Author(s)

This manuscript reports different common synthetic strategies for preparation of Prussian Blue Analogues (PBAs), particularly taking examples of Cu-Fe PBAs. Altogether 11 different Cu-Fe PBAs and one Mn-Fe based PBA have synthesized and characterized. The work is a comprehensive work on the synthetic strategies of PBAs and the manuscript provides various information on PBAs, which could be very helpful to material scientists working on PB structure-based materials. The manuscript is well-written and organized. In my views, the manuscript is suitable to be accepted in Royal Society Open Science after minor revision on the followings.

- (1) In addition to battery, PBAs have also been widely studied as electrocatalysts for water splitting and oxygen reduction reactions. This should be briefly discussed in the introduction part

and relevant reference should be cited. For example, *J. Mater. Chem. A*, 2016,4, 9781; *ACS Appl. Mater. Interfaces*, 2017 9, 18015, *J. Electroanal. Chem.*, 2018, 828, 80; *Nano energy*, 220, 68, 104371.

(2) In Page 11, Line 30: it is stated that “It is well known that Prussian White, FeII[FeII], easily oxidizes to Prussian Blue, FeIII[FeII]. In contrast, in Page11, Line 40-4: it is stated that “Mössbauer spectroscopy reveals partial reduction of assynthesized CuII[FeIII]: In Sample C, 34 % of iron has been reduced to iron(II),....”

Authors should explain why oxidation takes place selectively from FeII[FeII] to FeIII[FeII], but not to FeII [FeIII], while reduction take place from CuII[FeIII] to CuII[FeIII], but not to CuIII[FeII].

(3) JCPDS file number of CIF number of the standard reference PBAs added in Fig. S1 should be provide, which could be helpful for readers.

(4) In Fig. S2: (a) N is not shown; (e) and (f) Cu is not shown. Similarly, in Fig. S3: (g) N is not shown; (h) Cu is not shown. Please clarify, why distribution of those elements have not be shown.

Decision letter (RSOS-201779.R0)

Dear Professor Bentien:

Title: Strategies for Synthesis of Prussian Blue Analogues
Manuscript ID: RSOS-201779

Thank you for submitting the above manuscript to Royal Society Open Science. On behalf of the Editors and the Royal Society of Chemistry, I am pleased to inform you that your manuscript will be accepted for publication in Royal Society Open Science subject to minor revision in accordance with the referee suggestions. Please find the reviewers' comments at the end of this email.

The reviewers and handling editors have recommended publication, but also suggest some minor revisions to your manuscript. Therefore, I invite you to respond to the comments and revise your manuscript.

Because the schedule for publication is very tight, it is a condition of publication that you submit the revised version of your manuscript before 15-Nov-2020. Please note that the revision deadline will expire at 00.00am on this date. If you do not think you will be able to meet this date please let me know immediately.

When submitting your revised manuscript, you will be able to respond to the comments made by the referees and upload a file "Response to Referees" in "Section 6 - File Upload". You can use this to document any changes you make to the original manuscript. In order to expedite the

processing of the revised manuscript, please be as specific as possible in your response to the referees.

Kind regards,
Dr Laura Smith
Publishing Editor, Journals

On behalf of the Subject Editor Professor Anthony Stace and the Associate Editor Dr Dattatray Late.

RSC Associate Editor:
Comments to the Author:
Accept with minor revisions

RSC Subject Editor:
Comments to the Author:

(There are no comments.)

Reviewer comments to Author:

Reviewer: 1

Comments to the Author(s)

The paper entitled "Strategies for Synthesis of Prussian Blue Analogues" describes different syntheses and compares the structure of the obtained PBA materials. The PBAs are well characterized by PXRD, Mössbauer spectroscopy, and TEM studies. The manuscript is well organized and uncovered the properties of different PBAs. Therefore, the manuscript can be accepted for publication after minor revision.

The critical comments are as follow:

1. Grammatical errors should be checked again.
2. IR/Raman spectroscopy could be an important tool to understand the bridging of -CN group.
3. The PBAs prepared with potassium tetracyanonickelate should be discussed in the introduction.
4. Recent references for the application of PBAs (Mater. Today Energy 2020, 16, 100404, Chem. Asian J. 2020, 15, 607-623) should be included in the introduction part.
5. Author should describe how the tuneable structure of PBAs affects their electrochemical performance.
6. Author should explain the effect of the insertion of various metal ions at P and R sites in different PBAs.

Reviewer: 2

Comments to the Author(s)

This manuscript reports different common synthetic strategies for preparation of Prussian Blue Analogues (PBAs), particularly taking examples of Cu-Fe PBAs. Altogether 11 different Cu-Fe PBAs and one Mn-Fe based PBA have synthesized and characterized. The work is a comprehensive work on the synthetic strategies of PBAs and the manuscript provides various information on PBAs, which could be very helpful to material scientists working on PB structure-based materials. The manuscript is well-written and organized. In my views, the manuscript is suitable to be accepted in Royal Society Open Science after minor revision on the followings.

(1) In addition to battery, PBAs have also been widely studied as electrocatalysts for water splitting and oxygen reduction reactions. This should be briefly discussed in the introduction part and relevant reference should be cited. For example, J. Mater. Chem. A, 2016,4, 9781; ACS Appl. Mater. Interfaces, 2017 9, 18015, J. Electroanal. Chem., 2018, 828, 80; Nano energy, 220, 68, 104371.

(2) In Page 11, Line 30: it is stated that "It is well known that Prussian White, FeII[FeII], easily oxidizes to Prussian Blue, FeIII[FeII]. In contrast, in Page11, Line 40-4: it is stated that "Mössbauer spectroscopy reveals partial reduction of synthesized CuII[FeIII]: In Sample C, 34 % of iron has been reduced to iron(II),...."

Authors should explain why oxidation takes place selectively from FeII[FeII] to FeIII[FeII], but not to FeII [FeIII], while reduction take place from CuII[FeIII] to CuII[FeIII], but not to CuIII[FeII].

(3) JCPDS file number of CIF number of the standard reference PBAs added in Fig. S1 should be provide, which could be helpful for readers.

(4) In Fig. S2: (a) N is not shown; (e) and (f) Cu is not shown. Similarly, in Fig. S3: (g) N is not shown; (h) Cu is not shown. Please clarify, why distribution of those elements have not be shown.

Author's Response to Decision Letter for (RSOS-201779.R0)

See Appendix A.

Decision letter (RSOS-201779.R1)

Dear Professor Bentien:

Title: Strategies for Synthesis of Prussian Blue Analogues
Manuscript ID: RSOS-201779.R1

It is a pleasure to accept your manuscript in its current form for publication in Royal Society Open Science. The chemistry content of Royal Society Open Science is published in collaboration with the Royal Society of Chemistry.

On behalf of the Subject Editor Professor Anthony Stace and the Associate Editor Dr Dattatray Late.

RSC Associate Editor
Comments to the Author:
Accept as is

Reviewer(s)' Comments to Author:

Appendix A

Response to referees

Reviewer: 1

Comments to the Author(s)

The paper entitled "Strategies for Synthesis of Prussian Blue Analogues" describes different syntheses and compares the structure of the obtained PBA materials. The PBAs are well characterized by PXRD, Mössbauer spectroscopy, and TEM studies. The manuscript is well organized and uncovered the properties of different PBAs. Therefore, the manuscript can be accepted for publication after minor revision. The critical comments are as follow:

- 1) Grammatical errors should be checked again.
- 2) IR/Raman spectroscopy could be an important tool to understand the bridging of –CN group.

Answer: We agree, however, there is already substantial number of different experimental techniques involved (X-ray diffraction, Mössbauer spectroscopy, TEM and STEM-EDS) and we have in the present case decided to not to include more. Still, our conclusions hold and our work provide new information.

- 3) The PBAs prepared with potassium tetracyanonickelate should be discussed in the introduction.

Answer: PBA prepared from tetracyanonickelate yields a four-fold coordinated PBA, and is therefore structurally significantly different than PBA prepared from hexacyanoferrates. PBA offer such a wide range of structural tuning, and in the current study we focus on PBA prepared from hexacyanoferrate, stated on page 3, line 24.

- 4) Recent references for the application of PBAs (Mater. Today Energy 2020, 16, 100404, Chem. Asian J. 2020, 15, 607-623) should be included in the introduction part.

Answer: The references have now been included.

- 5) Author should describe how the tuneable structure of PBAs affects their electrochemical performance.

Answer: A section has been added to the introduction (end of page one to page two) on the relationship between structural variation and electrochemistry.

- 6) Author should explain the effect of the insertion of various metal ions at P and R sites in different PBAs.

Answer: We have now included this in the introduction (same section as indicated above).

Reviewer: 2

Comments to the Author(s)

This manuscript reports different common synthetic strategies for preparation of Prussian Blue Analogues (PBAs), particularly taking examples of Cu-Fe PBAs. Altogether 11 different Cu-Fe PBAs and one Mn-Fe based PBA have synthesized and characterized. The work is a comprehensive work on the synthetic strategies of PBAs and the manuscript provides various information on PBAs, which could be very helpful to material scientists working on PB structure-based materials. The manuscript is well-written and organized. In my views, the manuscript is suitable to be accepted in Royal Society Open Science after minor revision on the followings.

- 1) In addition to battery, PBAs have also been widely studied as electrocatalysts for water splitting and oxygen reduction reactions. This should be briefly discussed in the introduction part and relevant reference should be cited. For example, J. Mater. Chem. A , 2016,4, 9781; ACS Appl. Mater. Interfaces, 2017 9, 18015, J. Electroanal. Chem., 2018, 828, 80; Nano energy, 220, 68, 104371.

Answer: The references have now been included.

- 2) In Page 11, Line 30: it is stated that "It is well known that Prussian White, $\text{Fe}^{\text{I}}[\text{Fe}^{\text{I}}]$, easily oxidizes to Prussian Blue, $\text{Fe}^{\text{III}}[\text{Fe}^{\text{I}}]$. In contrast, in Page11, Line 40-4: it is stated that "Mössbauer spectroscopy reveals partial reduction of synthesized $\text{Cu}^{\text{I}}[\text{Fe}^{\text{III}}]$: In Sample C, 34 % of iron has been reduced to iron(II),...."
Authors should explain why oxidation takes place selectively from $\text{Fe}^{\text{I}}[\text{Fe}^{\text{I}}]$ to $\text{Fe}^{\text{III}}[\text{Fe}^{\text{I}}]$, but not to $\text{Fe}^{\text{II}}[\text{Fe}^{\text{III}}]$, while reduction take place from $\text{Cu}^{\text{II}}[\text{Fe}^{\text{III}}]$ to $\text{Cu}^{\text{I}}[\text{Fe}^{\text{III}}]$, but not to $\text{Cu}^{\text{III}}[\text{Fe}^{\text{I}}]$.

Answer: Copper(III) is not a common oxidation state for copper, and oxidation therefore takes place from $\text{Cu}^{\text{II}}[\text{Fe}^{\text{I}}]$ to $\text{Cu}^{\text{III}}[\text{Fe}^{\text{III}}]$.

In the case of iron hexacyanoferrate, Prussian Blue and Turnbull's blue were for a long time believed to be different compounds. Prussian Blue was prepared from an iron(III) salt and potassium ferricyanide, whereas Turnbull's Blue was prepared from an iron(II) salt and potassium ferrocyanide. The resulting compounds were believed to be $\text{Fe}^{\text{III}}[\text{Fe}^{\text{I}}]$ and $\text{Fe}^{\text{II}}[\text{Fe}^{\text{III}}]$, respectively. In 1968, Mössbauer studies proved that both compounds were identical, namely $\text{Fe}^{\text{III}}[\text{Fe}^{\text{I}}]$.¹

- 3) JCPDS file number of CIF number of the standard reference PBAs added in Fig. S1 should be provide, which could be helpful for readers.

Answer: The initial structural model was ICSD-89338. This is included in the experimental section under Materials characterization, Powder X-ray Diffraction.

- 4) In Fig. S2: (a) N is not shown; (e) and (f) Cu is not shown. Similarly, in Fig. S3: (g) N is not shown; (h) Cu is not shown. Please clarify, why distribution of those elements have not be shown.

Answer: Samples E, F and G are iron hexacyanoferrate (see table 1) and copper is therefore not present in the sample and is not included in STEM-EDS. For copper hexacyanoferrate samples the nitrogen distribution corresponds with the iron distribution (from $[\text{Fe}(\text{CN})_6]$ units, and for some copper hexacyanoferrate samples nitrogen is therefore not shown.

References

- 1 A. Ito, M. Suenaga and K. Ôno, *J. Chem. Phys.*, 1968, **48**, 3597–3599.